# Inferred Attractiveness: A generalized mechanism for sexual selection that can maintain variation in traits and preferences over time

Emily H. DuVal[1], Courtney L. Fitzpatrick[2], Elizabeth A. Hobson[3,4], Maria R. Servedio[5]*

**1** Department of Biological Sciences, Florida State University, Tallahassee, Florida, United States of America, **2** Department of Biology, Texas A&M University, College Station, Texas, United States of America, **3** Department of Biological Sciences, University of Cincinnati, Cincinnati, Ohio, United States of America, **4** Santa Fe Institute, Santa Fe, New Mexico, United States of America, **5** Department of Biology, University of North Carolina, Chapel Hill, North Carolina, United States of America

* ehduval@bio.fsu.edu (EHD), servedio@email.unc.edu (MRS)

## Abstract

Sexual selection by mate choice is a powerful force that can lead to evolutionary change, and models of why females choose particular mates are central to understanding its effects. Predominant mate choice theories assume preferences are determined solely by genetic inheritance, an assumption still lacking widespread support. Moreover, preferences often vary among individuals or populations, fail to correspond with conspicuous male traits, or change with context, patterns not predicted by dominant models. Here, we propose a new model that explains this mate choice complexity with one general hypothesized mechanism, "Inferred Attractiveness." In this model, females acquire mating preferences by observing others' choices and use context-dependent information to infer which traits are attractive. They learn to prefer the feature of a chosen male that most distinguishes him from other available males. Over generations, this process produces repeated population-level switches in preference and maintains male trait variation. When viability selection is strong, Inferred Attractiveness produces population-wide adaptive preferences superficially resembling "good genes." However, it results in widespread preference variation or nonadaptive preferences under other predictable circumstances. By casting the female brain as the central selective agent, Inferred Attractiveness captures novel and dynamic aspects of sexual selection and reconciles inconsistencies between mate choice theory and observed behavior.

## Introduction

Sexual selection occurs when individuals with certain phenotypes are disproportionately successful in obtaining mates and fertilizing gametes [1–3]. One powerful mechanism of sexual selection is mate choice, and coevolution of female preferences with male traits has long been

**Data Availability Statement:** The exact equations for the detailed model specification can be found in the Mathematica 12.0 code archived on the Open

Science Framework (DOI 10.17605/OSF.IO/R673J, which can be found on doi.org).

**Funding:** Financial support during preparation of this work was provided by National Science Foundation Integrative Organismal Systems award 1453408, Division of Biological Infrastructure award 1457541, and Division of Environmental Biology 2243423 (to EHD), National Institutes of Health T32 HD049336 (to CLF), National Science Foundation Integrative Organismal Systems award 2015932 and The Santa Fe Institute (to EAH), and National Science Foundation Division of Environmental Biology award 1939290 (to MRS). The funders had no role in study design, data collection and analysis, decision to publish, or preparation of the manuscript.

**Competing interests:** The authors have declared that no competing interests exist.

the primary focus of mate choice research [4,5]. However, mismatches between theoretical and empirical patterns demonstrate that mate choice models still lack critical components of the process by which preferences arise and are elaborated [6]. The dominant models of mate choice assume genetically inherited preferences but differ in the proposed mechanisms by which these preferences relate to fitness [7,8]. Under "good genes," females favor male traits that are honest indicators, e.g., of health or vigor [2,9]. In contrast, the runaway or "Fisherian" model proposes that preferences become exaggerated via genetic correlation with male traits, without additional fitness benefits [10–12]. Other hypotheses propose that an underlying sensory bias predisposes females to respond to particular traits [13,14], so that directional female preferences are maintained by processes outside of mate choice. The interplay between these models is a matter of ongoing debate [15], and such processes could work in concert as well as independently.

Each of these dominant mate choice models assumes genetic and trait-specific female preferences. However, preferences are widely influenced by female experience, including through copying, within-generational learning, and imprinting [16–18]. Preferences can also vary considerably among females in the same population [19,20], within a population over time [21], and among populations of the same species [22–24]. In situations where strong directional preferences do exist, it is a challenge to understand both how variation in male traits is maintained over time and why females devote considerable time and energy to choosing mates, issues that together comprise "the paradox of the lek" [25–28]. Here, we hypothesize a new mate choice process based on context-dependent social learning and model its effects over time. The proposed process leads to variation in preferences among females, rapid switching of population-level trait preferences, and maintenance of variation in male traits under many circumstances, recovering the dynamics of variation found in natural systems that have proven challenging to explain under previous mate choice models.

## The Inferred Attractiveness hypothesis

When mate choice occurs in a social context, females can obtain information about prospective mates by observing the choices of other females. However, when a male is chosen, it can be unclear why that individual was favored, as male phenotypic variation is complex and often multimodal [29–32]. The Inferred Attractiveness hypothesis proposes that females observe the mate choices of other females and compare a chosen male's phenotype to other available males to learn what phenotype best distinguishes him, in a form of negative frequency dependence (Fig 1A). In subsequent mate choice events, females choose mates that best match the learned template. Because female preferences in the population drive changes in male trait frequencies, the distinguishing trait variant eventually becomes more common. As the originally distinguishing trait (e.g., coloration in Fig 1A) becomes more common, it co-occurs with other variable traits (e.g., tail length in Fig 1A) that are alternative targets of preference. New observers may then (mistakenly) infer that an attractive male, chosen for the originally learned trait, is preferred for a different, more obviously distinctive trait. They then rapidly acquire preferences different from those of prior females.

The Inferred Attractiveness hypothesis makes 3 major modeling assumptions, which are all well supported by empirical studies: (1) social learning: females observe and change their behavior based on the choices of others [17,18,33]; (2) template formation: while observing matings, females generalize about the traits of chosen males, rather than learning to prefer specific individuals [33–36]; and (3) attentiveness to distinctive features: females use chosen males' most uncommon trait variants to categorize them as attractive, relative to other males [37]. Use of social information in mating need not have evolved originally in a mate choice

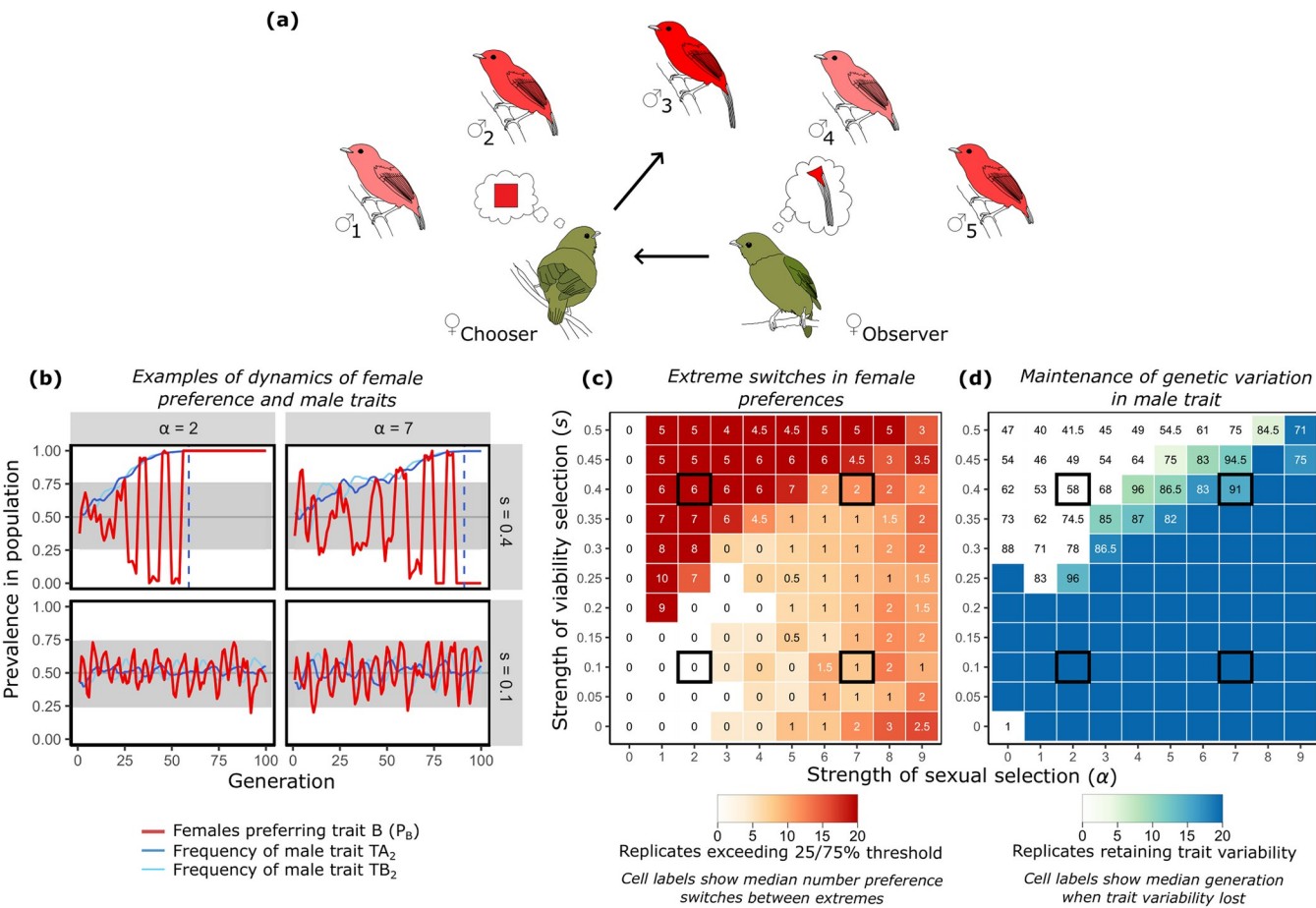

**Fig 1. Summary of the Inferred Attractiveness (IA) process and patterns.** Mate choice by IA is context dependent, influenced by both social information and relative rarity of courter traits. In hypothetical example **(a)**, the adult female choosing ($♀_{Chooser}$) prefers males with more saturated red plumage and, therefore, mates with $♂_3$. The juvenile female ($♀_{Observer}$) observes this choice without knowing which traits (color or tail length) determined $♀_{Chooser}$'s decision. Because tail length most noticeably distinguishes $♂_3$ from others, $♀_{Observer}$ infers long tails are attractive. Exemplar patterns produced by our model **(b**, at parameter values boxed in black in **c** and **d)** show that IA can produce variation over time in which trait is most commonly preferred (red line in **b**; in regions outside the shaded gray, ≥75% of females prefer the same trait). Population prevalence of female preference fluctuates, while the frequencies of male trait alleles (solid blue lines in **b**, showing adult trait frequencies censused after selection) vary. Strengths of viability selection (*s*) and female preference (*α*) interact to influence the prevalence of both **(c)** extreme fluctuations in which male trait is the focus of the female preference, and **(d)** variation in allele frequency of male traits, across 20 replicate simulations of the model. Dashed vertical lines in **(b)** indicate the generation in which trait variation is lost. X-axes of heat maps **(c)** and **(d)** show variation in *α*, where 1+ *α* represents how much more likely a female is to mate with a male that has her preferred trait variant. Y-axes show variation in viability selection, *s*, affecting each male trait, where a male with variant 1 is 1-*s* times less likely to survive than one with variant 2 of that trait. Here, we assume slight asymmetries (10%) in strengths across traits in both *s* and *α*. Code and raw simulation output used to generate this figure are archived on the Open Science Framework; DOI: 10.17605/OSF.IO/R673J.

context. Instead, such behavior could reflect a generalized tendency to use social information in decision-making, for example, in choice of foods [38], nesting [39,40] or oviposition [41] sites, and responses to predators [42].

Social learning, our first assumption above, was initially suggested as a factor influencing mate choice when researchers observed multiple females attending courtship displays of lekking male birds [43,44]. Nearly 2 decades later, laboratory trials demonstrated mate choice copying in guppies [45] and sparked similar tests in a range of taxa. Social information about potential partners influences mating decisions in many species [17,18], including humans [46,47]. Recent meta-analyses in nonhuman animals revealed that females who witness others choosing a male are on average 2.7 to 2.8 times more likely to choose that individual or a male

with similar traits [17,18], and effects are stronger when observers are sexually inexperienced [17]. Our second assumption, that observing females learn generalized information about preferred mates rather than copying choice of a specific individual, is supported by a subset of mate copying experiments that explicitly tested this possibility. In these studies, females received social information about males with distinct traits and then were given a choice between 2 new males differing in that specific noticeable trait [33,34]. Observer females in these experiments preferred the male with the distinctive phenotype of the previously chosen male. When compared across studies, the strength of generalized preference is indistinguishable from that of copying in individual-based experiments [18]. Finally, our third assumption, that females attend to distinctive traits when they learn that certain male phenotypes are attractive, is rooted in studies of discrimination learning and category learning [48,49]. Use of distinguishing features to define groups or classify items has been extensively demonstrated in humans [50–52] and in nonhuman animals [53,54]. Female preferences for "rare males" in some species indicate that free-living animals also utilize distinctive traits in mate choice [55–57]. The widespread phenomenon of habituation to commonly encountered stimuli provides a compelling mechanistic explanation for why rare features are distinctive [58]. These preferences typically manifest within the species' typical range of phenotypes, indicating that individuals making choices pay attention to distinctive stimuli that fall within acceptable limits.

## Model

To assess patterns of mate choice predicted by Inferred Attractiveness, we built a modified (phenogenotypic) population-genetic model with overlapping generations, explained in full in the Methods. Mating is polygynous and males have 2 genetically determined traits (TA and TB), each with 2 discrete trait variants (i.e., $TA_1$, $TA_2$, $TB_1$, and $TB_2$). Juvenile (first-year) observers see the mating choices of adult (second- and third-year) demonstrators and compare the chosen males to others in the population. Observers acquire a preference for the trait variant of the chosen male that is rarer in the population (i.e., the trait variant of the male that is more distinctive), out of the trait variants that he is expressing. They then later prefer this variant of that trait when choosing a mate. To explore the patterns generated by this process, we also vary the strengths of viability selection (selection via survival) on and preference for each trait and consider male traits that are environmentally rather than genetically determined. We model this process as genetically invariant across females.

## Results and discussion

The Inferred Attractiveness model produces repeated population-level switches in female preferences over time, both in which trait and in which trait variant are preferred (Fig 1B–1D). The relative trait frequencies are a fundamental underlying cause of these preference fluctuations: The identity of the most commonly preferred trait in the population tends to change when the 4 trait variants ($TA_1$, $TA_2$, $TB_1$, and $TB_2$) change rank order (i.e., when there is a change in the order of the frequency of the variants, from the most common to rarest in the population; Fig 2). As preferences shift to favor a relatively rare variant, sexual selection acts to increase the prevalence of this newly preferred variant, once again changing the rank order of the trait variants and causing a different variant to become relatively rare. This feedback perpetuates cycles of further fluctuations unless one of the 4 trait variants is lost (most commonly because of sufficiently strong viability selection favoring the alternate trait variant).

The interaction of sexual and viability selection determines patterns of trait as well as preference variation, via these effects on trait frequency. When directional viability selection is stronger than sexual selection (upper left quadrants, Fig 1B–1D), one variant of each trait type

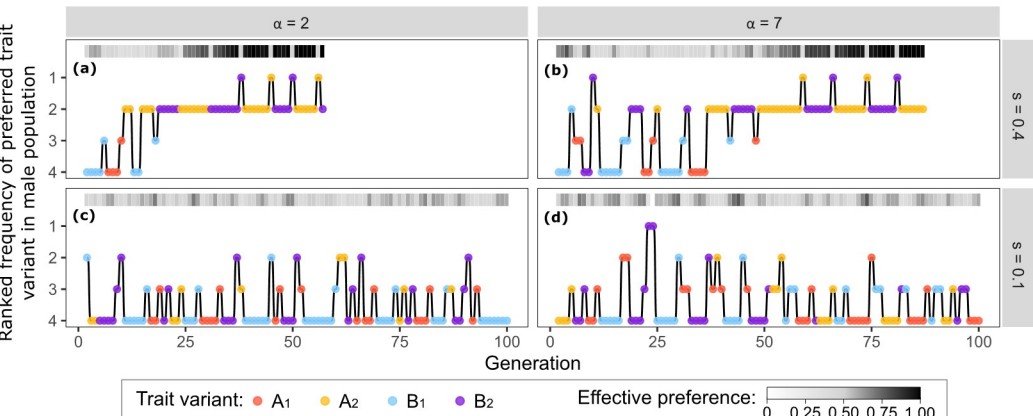

**Fig 2. Changes in rank order of variants underlie changes in population-level preference for traits and variants.** The relative frequency of the most widely preferred variant is shown here for different combinations of viability ($s$) and sexual ($\alpha$) selection. Data here are from a replicate of the Inferred Attractiveness model, different than that plotted in Fig 1B, and are archived with all code for generating figures as well as Mathematica code for the stochastic model at DOI 10.17605/ OSF.IO/R673J. Panels show example runs for generations 2–100, under **(a)** low sexual and high viability selection; **(b)** high sexual and high viability selection; **(c)** low sexual and low viability selection; and **(d)** high sexual and low viability selection (viability selection strength $s = 0.1$ or $0.4$; sexual selection strength $\alpha = 2$ or $7$). As in Fig 1, individuals with variant 1 of each trait are 1-s times less likely to survive than individuals with variant 2. Y-axes show the rank of the most preferred trait variant in the male population, from most common (rank 1) to rarest (rank 4). Point color indicates identity of the most preferred trait, and the greyscale at the top of panels indicates proportion of females expressing preference for this most commonly preferred trait variant. Code and raw simulation output used to generate this figure are archived on the Open Science Framework; DOI: 10.17605/OSF.IO/R673J.

tends towards fixation, and preferred trait variants tend to be those favored by viability selection. Females more frequently encounter these common trait variants and, therefore, tend to form preferences for the rarer of the 2 most common variants (most commonly the trait variant with rank 2, Fig 2; e.g., for a situation with trait variant frequencies $TA_1 = 0.3$, $TA_2 = 0.7$, $TB_1 = 0.1$, and $TB_2 = 0.9$, the population usually will exhibit preference for the rarer of the commonly encountered variants, here $TA_2$). This process produces large fluctuations in which trait is attended to by females, and these large preference fluctuations end only when one trait loses variation altogether (Fig 1C and 1D).

In contrast, when sexual selection is strong enough to counteract viability selection, the 4 variants remain more evenly distributed in the population (bottom right quadrants, Fig 1B–1D). Females then encounter all 4 variants at roughly the same rate and tend to form preferences for one of the 2 rarest trait variants (trait variants with ranks 3 and 4, Fig 2). Sexual selection will thus maintain variation in both traits over time (keeping variant frequencies close to 0.5, Fig 1B). In this scenario, preference fluctuations are dampened (i.e., females differ in which trait they prefer) and fluctuations are less regular, but last indefinitely within the bounds of drift. At preference strengths comparable to empirically measured mate choice copying effects (analogous to $\alpha = 2.5$ to 3, Fig 1) [17,18], trait variation is maintained even when viability selection exceeds levels typically observed [59]. The lack of consensus in preference that we find in this region is consistent with empirical patterns of variable mating preferences among females [20,21,60,61].

At the boundary between the regimes where viability selection or sexual selection dominates the behavior of the model, replicate runs show either the maintenance of trait variation during the timeframe of the simulations or the loss of variation in the trait. The simulations show evidence of a basin of attraction around a polymorphic state, but if the frequency of a trait variant that is favored by viability selection rises too high above this region, the frequency

of the favored variant increases until variation in this trait is lost (S3.ii Fig). We note that even when sexual selection is strong, the loss of variation at the trait loci is always theoretically possible given the stochasticity inherent in the models.

In all cases, the preference females develop for a trait variant makes that variant more common through sexual selection; as it increases in frequency, preferences shift away from that variant towards a different variant, causing preference fluctuations. Thus, preference acquisition via Inferred Attractiveness is not as simple as positive or negative frequency dependence. In order for a trait variant to become preferred by a female, she must observe another female mating with a male that has that variant (which is more likely it if is more common), but it must be the variant of that trait that is more unusual in the population (which means that it is the less common of observed options).

When strengths of sexual or viability selection differ across traits, as is commonly the case [13,14,62], the causes and consequences of changes in trait frequency rank and subsequent preference switching are entirely consistent with the mechanisms described in Figs 1 and 2. When viability selection is relatively weak on one trait, variants of the more selectively favored trait increase in frequency faster than do those of the less favored trait, so trait variants less often switch in relative frequency in the population (Fig 3Ai and 3Aiii). This results in fewer extreme switches in female preference over time (Fig 3Aii). When there are uneven strengths of sexual selection operating on each trait, which may be expected, e.g., if females have a sensory bias for one trait [5], resulting patterns of trait and preference variation are consistent with the mechanisms described above (Fig 3B). When viability selection is strong, weak sexual selection on one of 2 traits results in loss of trait variation at that locus, regardless of the strength of sexual selection on the other trait (upper portions of Fig 3Bi and 3Biii). As male trait variation trends towards loss (as in upper portions of Fig 3Bi), there are more occurrences of near-consensus of female preference in the population (i.e., large proportions of females preferring the same trait), and more occurrences of extreme switches in which trait is preferred (Fig 3Bi and 3bii).

Notably, our model predictions are very different from models of mate choice based on omniscient mate choice copying or frequency-dependent effects when copying does not occur (S1 Text; S1A and S1B Fig). When observer females learn the exact preference for the trait type targeted by the demonstrator (i.e., "omniscient" mate choice copying with no negative frequency dependence; S1A Fig), preferred trait variants generally become fixed in the population, and female preference does not tend to alternate among traits. When choice is instead frequency dependent due to encounter rate—similar to the IA model—but lacks social learning of preference (S1B Fig), preferences do not fluctuate over time and male trait variation is more frequently lost.

Results from Inferred Attractiveness are most similar to a model of preference for rare (or novel) males, with similarly overlapping generations, as both can produce extremes of female preference (i.e., large proportions of females exhibiting the same mating preference), maintain trait variation, and lead to repeated shifts in which trait is preferred (S3.xi and S2 Figs). When mate choice is based directly on rarity of male traits with no accompanying social learning of which males are attractive (a critical feature of Inferred Attractiveness), preferences fluctuate among traits in a manner similar to patterns generated by Inferred Attractiveness (S1C Fig). However, in this case, preference extremes are observed only in large groups (S3.xi versus S3.x Fig). In contrast, under Inferred Attractiveness, extremes of female preference can occur regardless of group size (S3.i–S3.iii Fig). Population extremes of preference are of interest as these are situations where an empirical measure of choice in the population would be likely to detect a correlation between male traits and reproductive success. When a preference for rarity or novelty directs mate choice in a large population, variation in male traits is maintained

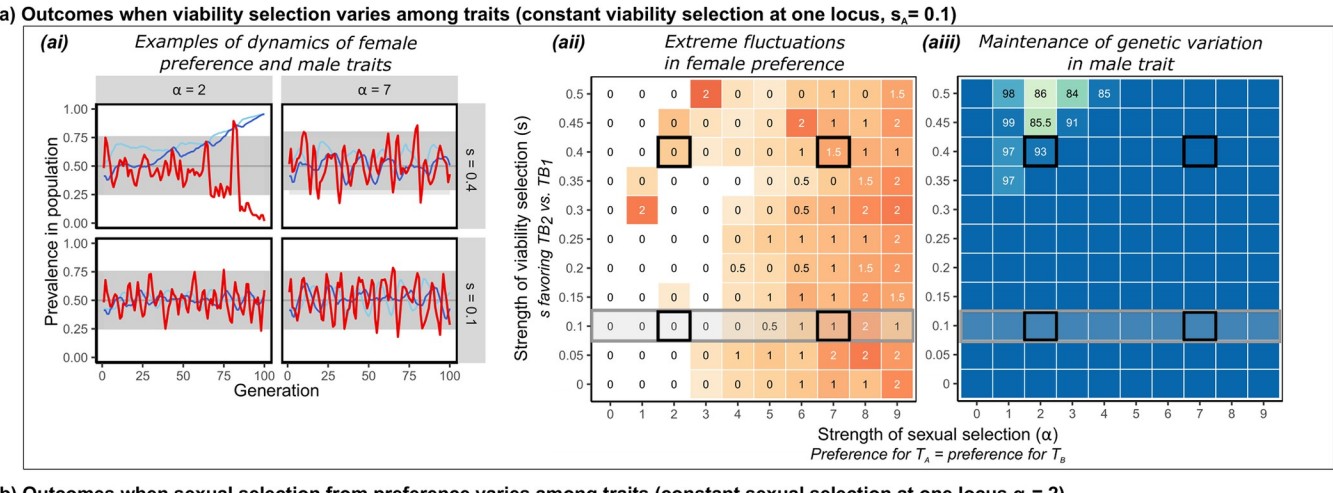

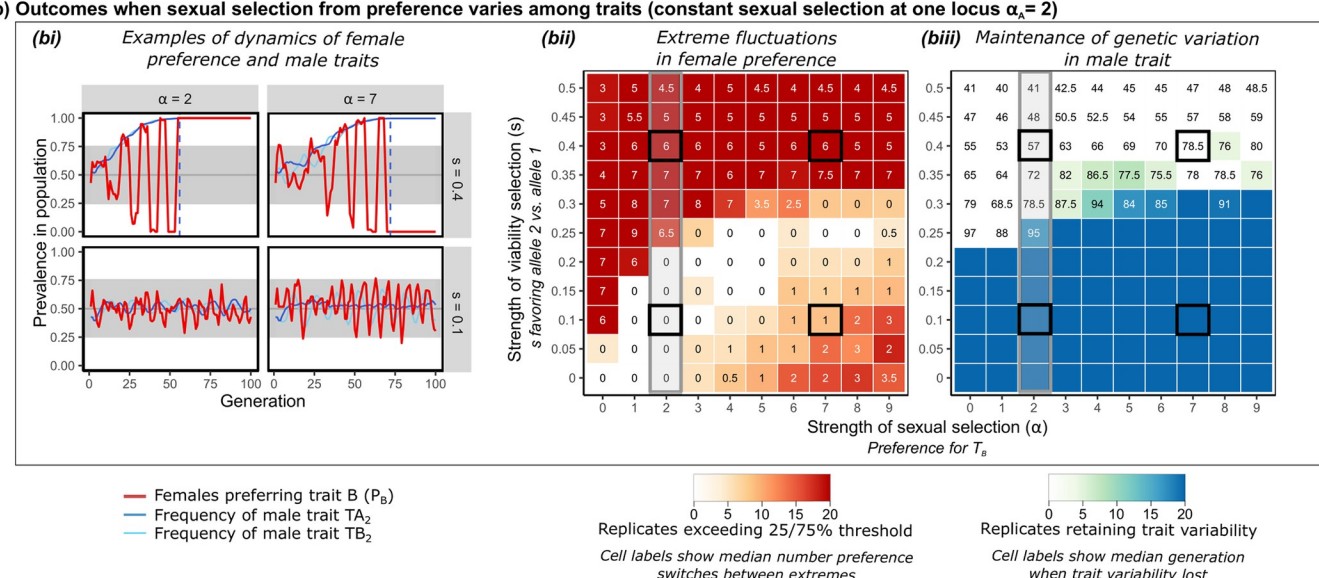

**Fig 3. Outcomes from Inferred Attractiveness when strength of selection (viability or sexual) differs between traits.** To assess the effects of varied viability selection **(a)**, we hold $s_A = 0.1$ and vary $s_B$ from 0 to 0.5 (where a male with variant 1 is 1-s times less likely to survive than one with variant 2 of that trait). The grey boxed area in **(aii)** and **(aiii)** indicates $s_A = s_B$; rows below this line therefore show outcomes when $s_A > s_B$, whereas rows above show outcomes when $s_A < s_B$. To assess effects of varied sexual selection **(b)**, we hold constant $\alpha_A = 2$ and vary $\alpha_B$ from 0 to 9 (where 1+ $\alpha$ represents how much more likely a female is to mate with a male that has her preferred trait variant). The area marked by the grey box in **(bii)** and **(biii)** indicates the region where $\alpha_A = \alpha_B$; columns to the left show outcomes where $\alpha_A > \alpha_B$, whereas columns to the right show outcomes when $\alpha_A < \alpha_B$. As before, dashed lines in plots of individual runs **(ai, bi)** indicate the generation in which trait variation is lost, or that trait variation persisted beyond the modeled timeline if the line occurs at generation 100. Plots **(ii)** and **(iii)** summarize results from 20 replicate runs of the stochastic model for each combination of parameter values, while plots **(i)** illustrate detailed patterns from one replicate at the parameter values indicated with black boxes in **(ii)** and **(iii)**. Exemplar plots in the bottom panels of **(ai)** and the left panels of **(bi)** therefore show the same parameter combinations as in Fig 1B (approximately equal s between traits in **ai**, and approximately equal $\alpha$ among traits in **bi**), albeit with data from a different replicate run of the model. X-axes of heat maps show variation in $\alpha$, where 1+$\alpha$ represents how much more likely a female is to mate with a male that has her preferred trait variant. Y-axes show variation in s affecting each male trait. Code and raw simulation output used to generate this figure are archived on the Open Science Framework; DOI: 10.17605/OSF.IO/R673J.

across nearly all tested scenarios of strength of sexual selection and viability selection (S1C Fig). A critical distinguishing outcome of the rarity model is that because females prefer the rarest available allele, they prefer the trait variants in our simulations that have a viability disadvantage (e.g., TA1 and TB1, S2 Fig). In contrast, Inferred Attractiveness is more likely to result in preference for and eventual fixation of viability-enhancing alleles (e.g., TA2 and TB2 in our simulations, Fig 2) when similarly strong viability selection is present (Fig 1 versus

S1C Fig). In other words, Inferred Attractiveness is more likely to produce female preference for trait variants that confer increased survival, a pattern that could be interpreted as "adaptive mate choice."

## Environmentally determined traits

The expression of some sexually selected traits is heavily influenced by environmental variance, which can change how complex signals evolve [63,64], especially when the environmentally determined traits co-occur with genetically determined traits. Fig 4 shows patterns produced by Inferred Attractiveness in a scenario in which the frequency of one trait (TB) is environmentally determined, with one variant occurring at a low frequency, 0.1. When sexual selection is weak relative to viability selection, the more viable genetic trait variant (TA$_2$) spreads rapidly in the population (upper left, Fig 4A). As this happens, the rank order of the trait variants can become fixed for long periods of time, producing widespread female preference for the environmentally determined trait that can last for many generations. In contrast, when sexual selection is strong relative to viability selection, it prevents a genetically determined trait from sweeping toward fixation (lower right, Fig 4A). Feedback then occurs between preference fluctuations and changes in rank order of trait variants, as described above. Such outcomes will depend on the frequency of the environmentally determined trait and are again consistent with the mechanisms explained above. Candidate traits where such effects may be important include behaviors flexibly expressed in different social or predation

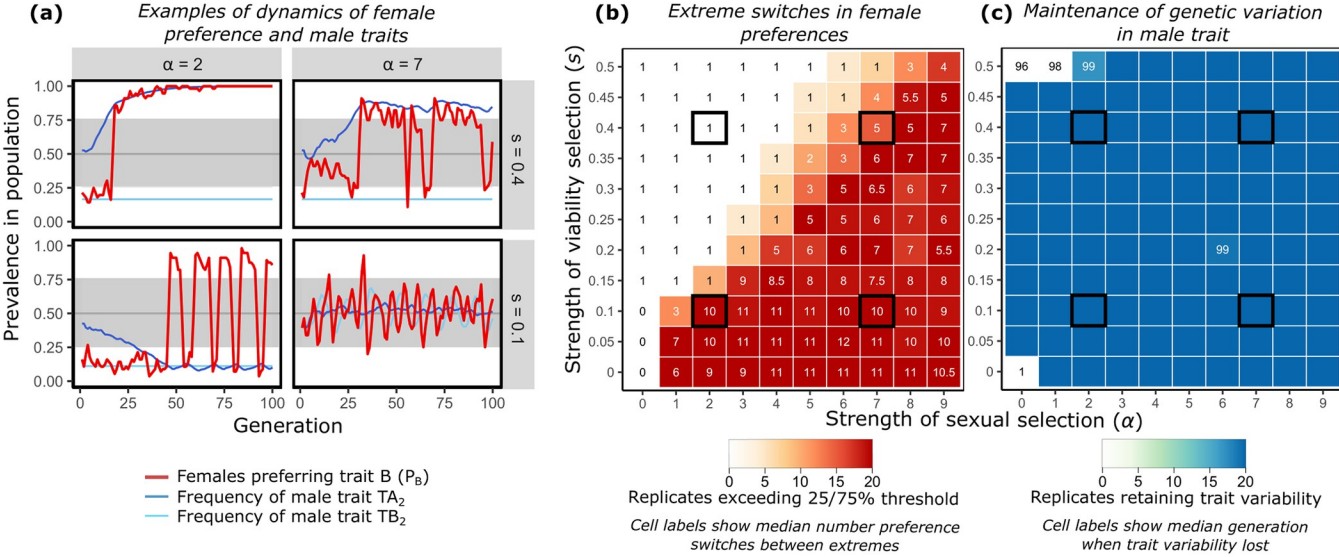

**Fig 4. Outcomes from Inferred Attractiveness in the presence of an environmentally determined trait.** Here, we model trait TB as environmentally determined and insensitive to effects of selection by resetting variant frequencies of TB in each generation (to TB$_2$ = 0.1), while variation in mating success and survival produce evolutionary change in frequency of the alleles at trait A. As in Figs 1 and 3, **(a)** shows exemplar patterns from single runs of the model at specified levels of α and s, with post-selection trait frequencies plotted. Panel **(b)** shows that repeated switches in population-wide female preferences are more common under certain circumstances, and **(c)** shows that variation in genetically determined male traits is maintained despite most changes in strengths of sexual and viability selection. When sexual selection is relatively weak (left region of each plot), near-uniform preference for the environmentally determined trait can occur for many generations. In contrast, when sexual selection is strong relative to viability selection on the genetic trait (bottom right regions), it can lead to the coincidence of trait frequencies. This causes females to switch from near-consensus in preference from one trait to the other as a previously more common trait variant becomes relatively rare. Replicate plots show that either the frequency of the favored or disfavored trait variant can become coincident with the frequency of the environmentally determined trait variant when viability selection is weak. Plot formatting follows that of previous figures. X-axes of heat maps show variation in α, where 1+α represents how much more likely a female is to mate with a male that has her preferred trait variant. Y-axes show variation in s affecting each male trait, where a male with variant 1 is 1-s times less likely to survive than one with variant 2 of that trait. Code and raw simulation output used to generate this figure are archived on the Open Science Framework; DOI: 10.17605/OSF.IO/R673J.

environments [65], or traits that vary with age [66]. The marked differences in pattern between this model (Fig 4) and the initial model (Fig 1) highlight the importance of understanding environmental or developmental effects on trait expression in predicting the evolutionary effects of Inferred Attractiveness.

## Implications and conclusions

Mate choice by Inferred Attractiveness produces sexual selection that can vary in strength and direction over time and generate rapid evolutionary changes in male phenotypes. It furthermore predicts maintenance of variation in both male traits and female preferences across generations under a wide range of circumstances. Inferred Attractiveness can generate transient patterns that, when measured on short time scales (<10 generations), conform to predictions of the current major models of mate choice. For example, under Inferred Attractiveness, there are circumstances when the majority of females prefer a trait that correlates with viability (as in good genes) and circumstances when the majority prefer a trait with no viability benefits— or indeed, that carry viability costs (as can occur in Fisherian "runaway" sexual selection). Variation in female preferences is a central prediction that emerges from the model—preferences may vary among females at one point in time (e.g., when sexual selection is the dominant force, bottom right, Fig 1), or they can vary over time (when viability selection overpowers sexual selection, top left, Fig 1). Finally, Inferred Attractiveness presents a plausible mechanism for female preferences that favor rare or novel males in some systems, even when such choices have no clear fitness effects [55,57,67]. The Inferred Attractiveness process is relevant for organisms that respond to social information, e.g., species ranging from fruit flies [33] to humans [47], and has implications for mate choice in hybrid zones [68] as well as among conspecifics.

Our model explicitly considers only a 2-trait system, but in practice, males exhibit many traits that may inform female mate choices. When males vary simultaneously in many traits, and when each trait encompasses multiple variants, we expect low consensus in female mating preferences to be the rule rather than the exception. This could also lead to sequential sexual selection on multiple traits over time and result in males with elaborated traits that are not necessarily the current targets of female choice. While our model does not depend on genetically determined female preferences, its proposed mechanism does rely on responses to stimuli that are mediated by female neurobiology. Brain structure and function itself has a genetic basis, and so perception and use of social information is influenced by genetic variation, and can evolve over time [69,70]. Indeed, evolved variation in neural response to different stimuli is a key component of interspecific behavioral differences, and innate and learned preferences can interact in complex ways [54,71]. These sensory biases will influence how female responses vary among the suite of available male traits and will define the range of perceptible traits to which females may respond [13,72]. Variation in learning processes could also influence expected outcomes of the Inferred Attractiveness process. For example, category learning (here, attractive or unattractive mates) can be influenced by relative variation in competing groups [73], and how categories are used can influence how they are learned [74]. For simplicity, our model also does not incorporate variation in the strength of social learning among females in the same population, but such variation is likely present.

The female brain is the selective agent of sexual selection by female mate choice. Inferred Attractiveness predicts variation both in female preferences and in male displays that is not present in other models of sexual selection. By directly incorporating female cognitive processes into choice of reproductive partners, Inferred Attractiveness highlights not only the power but also the flexibility of female mate choice as a mechanism of sexual selection.

## Methods

### Basic model

Using a population genetic framework, we model a system with female mate choice based on male traits. We refer to "female" choosers and polygynous "males" to facilitate comparison with existing models, but this terminology is analogous to the more general framework of "choosers" and "courters" [4], with the assumption that courters are mate limited and have variable mating success. Thus, this model would also apply to male-limited systems where males are the choosers. The exact equations for the detailed model below can be found in the Mathematica 12.0 [75] code archived on the Open Science Framework (DOI: 10.17605/OSF. IO/R673J). This archive also includes output from all simulated runs and complete code to the generate figures presented here.

We model a haploid system with 2 genetic loci, each with 2 alleles or phenotypes. These loci (TA and TB) control qualitatively different traits expressed only by males. For example, the TA locus might be a "color locus" where males can either have the allele to be light ($TA_1$) or dark ($TA_2$), while the TB locus might be a "pattern locus" where males can either be solid ($TB_1$) or have a striped pattern ($TB_2$). This produces 4 possible male genotypes (S1 Table; top row), each with a distinct male phenotype (S1 Table; middle row).

Females carry but do not express the alleles at these trait loci. Each female instead expresses 2 cultural traits that determine her preference, with 2 possible states each. These traits are acquired during the female's juvenile stage by observing mated pairs and drawing inferences—sometimes correctly and sometimes incorrectly—about which male traits drive the mating decisions of the observed adult females. The first cultural trait, P, determines whether a female will decide through this inference to base her preference on the TA locus or the TB locus; i.e., all females in our model have mate preferences, but some females base their preferences on male phenotypes at the TA locus ($P_A$ females) and some females base their preferences on male phenotypes at the TB locus ($P_B$ females). Once acquired, an individual female's preference does not change.

The second cultural trait expressed by the female (OA or OB) determines which trait variant she will prefer at the trait locus that she focuses on. When females base their preferences on expression at the TA locus (i.e., have the $P_A$ cultural trait variant), the corresponding cultural trait OA determines which of the 2 possible alleles at TA is preferred; females with the $OA_1$ cultural trait variant prefer males with the $TA_1$ allele (over $TA_2$), and, similarly, females with $OA_2$ prefer males with $TA_2$ (over $TA_1$). Likewise, when females base their preferences on expression at the TB locus (i.e., have the $P_B$ trait variant), the corresponding OB trait determines which of the 2 TB alleles are preferred (see S1 Table; bottom row). Note that for a $P_A$ female, the OB trait is not expressed, and for a $P_B$ female, the OA trait is not expressed (although we store trait variant information at these alternate traits for ease of model development; they are never used by females).

The combination of genetic loci and cultural trait phenotypes are termed "phenogenotypes" in the literature on gene-culture coevolutionary theory (reviewed in [76]). In our model, adult females have information stored at 3 cultural trait phenotypes (including the unexpressed but still-stored cultural trait) and at 2 unexpressed trait loci, or at 5 positions in total, producing 32 possible phenogenotypes, $P_A OA_1 OB_1 TA_1 TB_1$, $P_A OA_1 OB_1 TA_1 TB_2$, $P_A OA_1 OB_1 TA_2 TB_1$,... $P_B OA_2 OB_2 TA_2 TB_2$. Because female zygotes have not yet observed adult matings (which occur during the juvenile stage as specified below), they carry only alleles at the 2 display trait loci, which are not expressed in females. Likewise, adult males, which do not express or store information about preferences, also carry only alleles at the 2 trait loci, producing 4 possible genotypes. In both cases, these genotypes are ordered as $TA_1 TB_1$, $TA_1 TB_2$, $TA_2 TB_1$, and $TA_2 TB_2$.

Because juvenile females learn from adults, a model of Inferred Attractiveness must contain overlapping generations and multiple age cohorts. In our model implementation, we specifically track 3 age classes that are all present in the population at any given time: (1) young adults; (2) older adults; and (3) juveniles (which, as described above, have fewer phenogenotypes to track since they have not yet acquired their preferences). Each age cohort lasts only for 1 year. Young adults are represented by phenogenotype frequencies $x_{f,1}$ through $x_{f,32}$ at time $t$ for females (which sum to 1) and $x_{m,1}$ through $x_{m,4}$ at time $t$ for males (also summing to 1), where the phenogenotypes and genotypes are ordered as specified in the paragraph above. Older adults are represented by phenogenotype frequencies $x_{f,33}$ through $x_{f,64}$ at time $t$ for females (summing to 1, these are equivalent to $x_{f,1}$ through $x_{f,32}$ at time $t–1$) and $x_{m,5}$ through $x_{m,8}$ at time $t$ for males (summing to 1, equivalent to $x_{m,1}$ through $x_{m,4}$ at time $t–1$ after the males have undergone an additional round of viability selection based on their traits). Young and older adult males both experience viability selection based upon their trait phenotype and then proceed through a mating step, where mating occurs across both adult cohorts following the assumptions of polygyny (see below). These adult matings are observed by the juveniles (represented by genotype frequencies $x_{f,65}$ through $x_{f,68}$ at time $t$ for females and $x_{m,9}$ through $x_{m,12}$ at time $t$ for males, again both of these sum to 1). These juveniles thus consist of the offspring from individuals who were adults (of any age) at time $t–1$. Through these observations, juvenile females obtain their full phenogenotypes (with mate preferences) and become young adults at time $t+1$. Juvenile males at time $t$ also become young adults at time $t+1$ but do not acquire mating preferences.

## Viability selection

Females do not undergo differential viability selection because they do not express TA and TB. Thus, after the viability selection step of the life cycle, female frequencies are denoted $x_{f,i}^{vs}$ (where $x_{f,i}^{vs} = x_{f,i}$, and $i$ indexes the phenogenotypes 1 through 32 in young adults and 33 through 64 in older adults in females). In contrast, we assume that males undergo differential survival both as young adults and older adults (but not as juveniles, which we assume do not yet express display traits), based upon expression at the TA and TB loci. In particular, alleles $TA_2$ and $TB_2$ are favored by viability selection, where the selection coefficients on male traits are represented by $s_a$ (yielding fitness $1–s_a$ for allele $TA_1$ and 1 for allele $TA_2$) and $s_b$ (yielding fitness $1–s_b$ for allele $TB_1$ and 1 for allele $TB_2$). The frequencies of male genotypes after viability selection are as follows (where $j$ indexes phenogenotypes 1 to 4 in young adults and 5 to 8 in older adults in males, and the denominator normalizes the frequencies by summing the product of the fitnesses and frequencies, over all genotypes $k$):

$$x_{m,j}^{vs} = \frac{(1 - d_{Aj}s_a)(1 - d_{Bj}s_b)x_{m,j}}{\sum_k (1 - d_{Ak}s_a)(1 - d_{Bk}s_b)x_{m,k}}. \tag{1}$$

Here, $d_{Aj} = 1$ when $j \bmod 4 = 1$ or 2 (when adult males carry the $TA_1$ allele; "$j \bmod 4$" refers to the remainder when $j$ is divided by 4) and otherwise, $d_{Aj} = 0$, and similarly $d_{Bj} = 1$ when $j \bmod 4 = 1$ or 3, (when adult males carry the $TB_1$ allele), and, otherwise, $d_{Bj} = 0$, where $j < 9$ (selection occurs in adults but not juveniles, for which $9 \leq j$ and $x_{m,j}^{vs} = x_{m,j}$). Note that Eq (1) applies separately for young adults (with genotypes $j = 1$ through 4) and older adults (with $j = 5$ through 8), causing the sums of the genotypes $x_{m,j}^{vs}$ of each age cohort to equal 1 after selection.

After viability selection, females choose mates from among all adult males, regardless of their age. To accomplish this in the model, male genotype frequencies are combined as weighted averages into one male mating pool. Specifically, the frequencies of the alleles in each

adult cohort of males are weighted by the mean survivorship of that cohort, to account for the fact that males of the older cohort have undergone 2 bouts of viability selection (i.e., there are fewer of them than males of the younger cohort). Our model considers age-related differences in the tendency of young females to learn from the mate choices of others [17], but we note that effects of age in natural populations also include nuances not incorporated here. For example, female fecundity scales positively with age in some systems [77], as does male ornamentation and reproductive success [78,79]. Thus, after viability selection and averaging across the young and older adult cohorts, male genotype frequencies are represented for a given time step $t$ as

$$\bar{x}_{m,j}^{vs}(t) = \frac{x_{m,j}^{vs}(t)\bar{w}_{AY}^{v}(t) + x_{m,j+4}^{vs}(t)\bar{w}_{AO}^{v}(t)\bar{w}_{AY}^{v}(t-1)}{\bar{w}_{AY}^{v}(t) + \bar{w}_{AO}^{v}(t)\bar{w}_{AY}^{v}(t-1)}. \tag{2}$$

Here, as $j$ goes from 1 to 4, first the frequencies of young males with a given genotype are weighted by mean survivorship (mean fitness through viability selection alone, $\bar{w}_{AY}^{v}(t)$, at the current time step). Second, the frequencies of older males of that same genotype are weighted both by mean survivorship ($\bar{w}_{AO}^{v}(t)$, in the current time step for their second bout of selection, when older) and by their mean survivorship in the previous time step ($\bar{w}_{AY}^{v}(t-1)$, their first bout of selection, when they were young adults). The expressions are normalized to ensure that they are maintained as frequencies. Note that Eq (2) is valid regardless of whether the fitnesses 1, $1-s_a$, and $1-s_b$ are absolute or relative fitnesses. Note also that Eq (2) relies on the assumption that there are always the same number of zygotes produced each generation. This is warranted because females are not under selection, so the number of females can be assumed to be unchanged through each generation.

## Mating and sexual selection

After viability selection, mating takes place under strict polygyny. All females make their mating decisions based on only one of the 2 traits; females with the $P_A$ cultural trait base their preferences on the phenotype they observe at the male's TA locus (e.g., color), and females with the $P_B$ cultural trait base their preferences on the phenotype at the male's TB locus (e.g., pattern). These preferences are culturally transmitted through observations when the females are juveniles (see Observations section below). Likewise, females obtain a culturally transmitted preference for one of the 2 possible variants at each locus (these preferences are stored at the OA and OB phenotypic positions in the phenogenotypes). The strength of the mating preference is either denoted by $\alpha_a$ (for $P_A$ females) or by $\alpha_b$ (for $P_B$ females), where the preference strength $1+\alpha_a$ or $1+\alpha_b$ is defined by how much more likely a female is to mate with a male that has the trait variant she prefers than a male that does not, if she were to encounter one of each.

The implementation of these assumptions in our model is as follows. In order to calculate the frequencies of mated pairs of different phenogenotypes, first we create a matrix of mating preferences **M** with 32 rows (for all female phenogenotypes $i$) and 4 columns (for the male genotypes $j$, see S2 Table). Specifically, the elements of **M** represent the relative mating preferences of a female if she encounters one male of each genotype, such that

$$M_{ij} = (1 + g_{Aij}\alpha_a)(1 + g_{Bij}\alpha_b). \tag{3}$$

Here, $g_A$ and $g_B$ identify whether a female is basing her preferences on trait TA or TB, such that $g_{Aij} = 1$ when $i$ is 1 to 8 and $j$ is 1 or 2 (i.e., females are $P_AOA_1$ and males are $TA_1$), or if $i$ is 9 to 16 and $j$ is 3 or 4 (i.e., females are $P_AOA_2$ and males are $TA_2$), and otherwise $g_{Aij} = 0$. Similarly, $g_{Bij} = 1$ when $i$ is 17 to 20 or 25 to 28 and $j$ is odd (i.e., females are $P_BOB_1$ and males are

TB$_1$) or if $i$ is 21 to 24 or 29 to 32 and $j$ is even (i.e., females are P$_B$OB$_2$ and males are TB$_2$) and otherwise $g_{Bij} = 0$.

As in many sexual selection models, females mate nonrandomly according to their mating preferences, weighted by their probabilities of encounter with each male genotype and with the constraint that each female has equal mating success. Specifically, young adult females ($x^{vs}_{f,1}$ through $x^{vs}_{f,32}$) and older adult females ($x^{vs}_{f,33}$ through $x^{vs}_{f,64}$) mate nonrandomly across the 4 male genotypes whose frequencies across age classes have been calculated from Eq (2), $\bar{x}^{vs}_{m,1}$ through $\bar{x}^{vs}_{m,4}$, producing the 32 × 4 matrices $\mathbf{F_{AY}}$ (mating table for young adult females) and $\mathbf{F_{AO}}$ (mating table for older adult females). The elements in each of these matrices represent the frequencies of random encounters between a female of phenogenotype $i$ and a male of genotype $j$, scaled by the mating preferences $M_{ij}$ to produce the relative proportion of the population that consists of each type of mated pair. Thus,

$$F_{AYij} = \frac{M_{ij}x^{vs}_{f,i}\bar{x}^{vs}_{m,j}}{\sum_k M_{ik}\bar{x}^{vs}_{m,k}} \text{ and } F_{AOij} = \frac{M_{ij}x^{vs}_{f,i+32}\bar{x}^{vs}_{m,j}}{\sum_k M_{ik}\bar{x}^{vs}_{m,k}}. \tag{4}$$

The denominators of the expressions in (4) assure that each adult female, regardless of her age, has equal mating success, producing a mating system where only males are under sexual selection.

## Zygote formation

Because only the traits TA and TB are genetically inherited, zygote formation produces individuals that are distinguished by these 2 loci but not by the cultural traits that we track above (P, OA, and OB). For simplicity, we assume that free recombination takes place between these 2 genetic loci. In order to account for recombination, we collapse the elements of $\mathbf{F_{AY}}$ and $\mathbf{F_{AO}}$ to create matrices that depend only on genotype frequencies at the TA and TB loci (irrespective of the P, OA, and OB phenotypes). The resulting 4 × 4 matrices $\mathbf{G_{AY}}$ and $\mathbf{G_{AO}}$ are for young and older adult females, respectively. Zygote formation is calculated by summing the appropriate elements from the 2 mating matrices, accounting for recombination and segregation, following the standard equations for recombination and segregation in haploids. This results in 4 types of zygotes, which at this age do not yet have mating phenotypes, but only genotypes at the TA and TB loci. Zygotes are created separately from young and older adult females and then averaged to produce a final zygote pool (since there are equal numbers of females in each age cohort). We assume zygotes have an equal sex ratio and are indexed in females as phenogenotype frequencies $x_{f,65}$ through $x_{f,68}$ and males as genotype frequencies $x_{m,9}$ through $x_{m,12}$ (as specified above).

## Female observations and acquisition of preferences

Our primary objective is to examine the temporal dynamics of female mate preferences and male traits when female mating preferences are both culturally transmitted and context dependent. Thus, our hypothesized phenomenon represents an interaction between 2 processes that ultimately give rise to female mating decisions. First, juvenile females acquire their mating preferences from adult females in the population (cultural transmission). Second, this cultural transmission of mating preferences is not simply a direct transmission of identical preferences from female to female but, instead, can change as a function of the distribution of male traits in the population. In other words, there is frequency dependence.

Below, we describe the acquisition of preferences under the Inferred Attractiveness hypothesis; "Reference Models" that isolate assumptions of the model, constructed for comparison,

are included in the Supporting Information (S1 Text and S1 Fig). All of these models (IA and the Reference Models) involve stochasticity in the form of limited numbers of females observing relatively small numbers of males, sampled from the entire population. These are realistic features of study populations targeted by empiricists and so are important to include. However, we wanted to limit the effects of stochasticity to its role in setting preferences, as opposed to introducing stochastic changes in trait frequencies (which would obfuscate the effects of the preferences on trait evolution). We therefore renormalize the phenogenotype frequencies after preferences are set, in a way that preserves both the trait frequencies from before preference acquisition (in the females; these are unaffected in the males since males do not acquire preferences) and that also preserves any statistical associations (linkage disequilibrium) that has formed between the traits. This is done by first summing across the 2 genetic trait loci to calculate the frequency of each female preference phenotype and then distributing these preference phenotype frequencies evenly across the genetic trait frequencies that are carried by females entering the mating step of the life cycle.

## Inferred Attractiveness model: Context-dependent cultural transmission of mate preferences

In the Inferred Attractiveness model, female decisions about mating are influenced both by social information (mate choice copying) and by the distribution of traits in the population (i.e., mating decisions follow a set of frequency dependent rules). Recall that the cultural trait P determines whether females pay attention to male trait TA (e.g., color; females with the $P_A$ cultural trait variant) or to male trait TB (e.g., pattern; females with the $P_B$ cultural trait variant). Specifically, we assume that juvenile females set their cultural trait variant at P ($P_A$ or $P_B$) by observing an adult female mating and inferring that she is basing her choice on the male's phenotype that is the most unusual in the population and thus most distinctive in that male (out of $TA_1$, $TA_2$, $TB_1$, and $TB_2$). If the juvenile infers that the observed female is basing her choice on one of the alleles at the TA locus, for example, the juvenile develops phenotype $P_A$, meaning she pays attention to TA when she is making a mating decision (likewise inferred observations of TB will set the phenotype $P_B$). The juvenile will then prefer the trait variant that the observed female has chosen at the relevant trait locus. For example, if a juvenile develops phenotype $P_A$, and the observed female was mating with a male with trait $TA_1$, the juvenile will also prefer $TA_1$ males (e.g., she will develop phenotype $OA_1$).

   Operationally, we simulate a finite number of juvenile females (a relatively small local population), each observing a single successful mating in proportion to how frequently (by male genotype) that type of mating occurs. A set number of females of each phenogenotype $i$, $x_{f,i}^N$, is modeled by rounding down from $n\,x_{f,i}^{vs}$, where $n$ is roughly the number of females in the local population (the actual local population size of females, $\sum_i x_{f,i}^N$, is somewhat smaller due to the rounding described earlier in this sentence; the superscript $N$ indicates a number of females rather than a frequency). To simulate each of these females randomly observing a successful mating, a random number between 0 and 1 is chosen for each juvenile female and matched to the genotype of a male by assigning bins for each type of male between 0 and 1 in proportion to their frequency among successfully mated males. The genotype of the successfully mating male is then assessed to determine whether the allele that he carries at the TA locus or at the TB locus is rarer in the population at large. If the allele at the TA locus is rarer, the juvenile female will acquire phenotype $P_A$, and her OA phenotype will be set to $OA_1$ if the male is $TA_1$ and $OA_2$ if the male is $TA_2$. Phenotypes at the P and OB cultural traits will be set analogously if the allele that the observed male carries at the TB locus is rarer in the population than the one

that he carries at the TA locus. Note that we store phenotypes at both the OA and OB trait for every observation, even though we only use the OA phenotype if the female is $P_A$ and the OB phenotype if the females is $P_B$, as described above, to allow future flexibility in the operation of the code.

## Inferred Attractiveness with an environmentally determined male trait

Some display traits in nature may vary substantially in expression with environmental conditions or age. We explore this possibility by altering our assumptions to investigate the effects of environmentally determined male traits. In this scenario, the population frequency of one of the male traits before viability selection does not change across generations. This could occur, for example, if variation in a trait relevant for mate choice is determined by resource availability at key life stages [80], by labile responses to predators or social contexts [81], or by climactic variation [63,82]. Furthermore, age-based variation in sexually selected traits is common [83].

To investigate environmentally determined traits, we assumed that the TA locus remains genetically determined and varies in frequencies among generations, as above. However, the phenotypes at the TB locus remain at a constant frequency in zygotes across generations, as determined by the environment. We model this by allowing all genotypes to proceed exactly as described for the main model above. However, the zygote genotype frequencies in each generation are recalculated to reset the allele frequencies at the TB locus to their starting values.

## Supporting information

**S1 Text. Reference models for comparison to the Inferred Attractiveness model.** To investigate how the interaction of cultural transmission and context dependence differs from either such circumstance in isolation, we include, in addition to our basic model, 3 reference models that consider these 2 processes (cultural transmission and frequency dependence) independently. These are Reference Model 1: Direct Copying of Female Preferences (Cultural Transmission), Reference Model 2: Preference for Male Trait that has the Most Rare Variant, and Preference for a Trait Variant from a Randomly Observed Male (IA without social learning), and Reference Model 3: Preference for Rare/Novel Male Traits and Trait Variants. S1 Text explains the parameters of and major findings from each reference model, and results are shown graphically in S1 Fig.
(DOCX)

**S1 Fig. Patterns generated by comparison models demonstrate the ways in which Inferred Attractiveness differs from similar scenarios.** In each case, (i) panels show dynamics of preference and trait variation over time for one exemplar of 20 replicate runs (at parameter values highlighted by bold black boxes in the other plots). Heat maps (panels ii and iii) summarize trends from 20 replicate runs of the model using the conventions established in the figures of the main text. As in previous plots, dashed lines in plots of individual runs (i) indicate the generation in which trait variation is lost. X-axes in (ii) and (iii) define the strength of female preference ($\alpha$), which is how much more likely a female is to mate with a male that has the trait variant she prefers, relative to a male with the unpreferred variant at that same locus and given that she encounters one of each type. The y-axes in (ii) and (iii) represent the strength of viability selection ($s$) on allele 2 of each male trait, such that a male with variant 1 is 1-$s$ times less likely to survive than a male with variant 2 of the same trait. Code and raw simulation output used to generate this figure are archived on the Open Science Framework; DOI: 10.17605/OSF.IO/R673J.
(TIFF)

**S2 Fig. Changes in rank order of preferred trait variants generated by the preference for rarity model (S1C Fig) show patterns distinct from those generated by Inferred Attractiveness (Fig 2).** Even when selection favoring some trait variants is extremely strong, females making mate choices by rarity or novelty preference prefer the rare (selectively disfavored) variants. In comparison, mate choice by inferred attractiveness relatively quickly favors selectively advantageous variants when selection is strong (Fig 2). Therefore, although both preference for rarity and IA produce fluctuations in which trait is preferred and maintain variability in mate traits over time, IA does so in a manner that is more responsive to variation in selective strength, via changes in the relative frequency of trait variants. Under a preference for rarity, female preference targets selectively disadvantageous trait variants ($A_1$ or $B_1$) when viability selection on those traits is strong. Panels show generations 2–100 for (a) low sexual selection and high viability selection; (b) high sexual selection and high viability selection; (c) low sexual selection and low viability selection; and (d) high sexual selection and low viability selection (sexual selection strength 0.2 or 0.8; viability selection strength 2 or 7). Y-axes show how common the variants are in the male population ranked from most common (rank 1) to rarest (rank 4), point color indicates which trait variant is most preferred, and the greyscale at the top of the panels indicates the proportion of females expressing preference for the most commonly preferred trait variant. Strength of sexual selection here reflects female preference ($\alpha$), defined as above as how much more likely a female is to mate with a male that has the trait variant she prefers. Code and raw simulation output used to generate this figure are archived on the Open Science Framework; DOI: 10.17605/OSF.IO/R673J.
(TIFF)

**S3 Fig. Exemplar model runs showing example output from a single panel of runs of each model discussed.** Plots show details from the full considered range of viability selection and preference strength. Output on each page represents one run of each model, and models are identified by the combination of model, setting, and group size indicated in the header on each page. Code and raw simulation output used to generate these figures are archived on the Open Science Framework; DOI: 10.17605/OSF.IO/R673J. The first page of the set of figures provides a key identifying the parameters used in each model shown. Most models are shown with male group size 30, which we considered the most biologically realistic, but we present the main model at smaller (S3.i) and larger (S3.iii) male group sizes to illustrate the effects of group size on outcomes. In general, when IA occurs in larger male group sizes, stochasticity of outcomes is reduced. We also present outcomes for model pnov.x at the larger group size (100 males, S3.xi), as this version of the model produced larger fluctuations in female preference than observed at smaller male group sizes. Grey bars along the top of each plot grid indicate the level of sexual selection ($\alpha$) for subplots in that column, while grey bars along the left side of plots indicate the level of viability selection for subplots in that row. X-axes of each subplot indicate generation, and y-axes represent frequency. Red lines plot the frequency of female preference for trait TB ($p_B$, where $1-p_B$ females prefer trait TA); dark blue lines indicate the frequency of trait variant $TA_2$, and light blue lines indicate frequency of $TB_2$, with frequencies plotted after selection in each generation. Horizontal lines indicate a frequency of 0.5 for reference. Text in the upper left of each subplot indicates the last generation in which traits were variable (such that gen. fixed = 100 indicates that traits maintained variation in all plotted generations), and this generation is also indicated by a vertical dashed blue line. In S3 plots, if trait variation persisted beyond the modeled timeline, the dashed line occurs at generation 100. Text in the lower right indicate the number of extreme switches, defined as situations where >75% of females in a population prefer one trait, but, later, >75% of females in the population

prefer the other trait. Two consecutive switches constitute a "fluctuation."
(PDF)

**S1 Table. Model specification of male genotypes, phenotypes, and corresponding phenotypes of females that prefer these males.** The mating decisions of females with the $P_A$ phenotype are influenced by male trait locus TA (here, color) and the mating decisions of females with the $P_B$ phenotype are influenced by male trait locus TB (here, pattern). The OA and OB cultural traits carry preference phenotypes for one of the alternate male alleles (1 or 2) of each type of trait (e.g., for the "striped" phenotype at the "pattern" locus), and females acquire variant information for both TA and TB from the chosen male genotype. As females attend to only one trait locus at a time, the phenotype at only one of the OA or OB positions is expressed (indicated in bold).
(DOCX)

**S2 Table. Coefficients determining nonrandom mating (the numerators in Eqs (2) and (3)) across all combinations of mated genotype pairs in the Inferred Attractiveness model.** Subscripts 1–32 are for young adult females and subscripts 33–64 are for older adult females.
(DOCX)

## Acknowledgments

H. L. Anderson, T. Aubier, D. Houle, M. Kuzel, E.A. Lacey, B. Lerch, P. Rivers, J. Travis, and K. Xu provided helpful comments and discussion during development of these ideas. This research team benefitted from 2 working group meetings at the Santa Fe Institute.

## Author Contributions

**Conceptualization:** Emily H. DuVal.

**Data curation:** Courtney L. Fitzpatrick, Elizabeth A. Hobson, Maria R. Servedio.

**Formal analysis:** Emily H. DuVal, Courtney L. Fitzpatrick, Elizabeth A. Hobson, Maria R. Servedio.

**Funding acquisition:** Emily H. DuVal, Elizabeth A. Hobson.

**Investigation:** Emily H. DuVal, Courtney L. Fitzpatrick, Elizabeth A. Hobson, Maria R. Servedio.

**Methodology:** Emily H. DuVal, Courtney L. Fitzpatrick, Elizabeth A. Hobson, Maria R. Servedio.

**Project administration:** Emily H. DuVal.

**Supervision:** Emily H. DuVal, Maria R. Servedio.

**Validation:** Courtney L. Fitzpatrick, Elizabeth A. Hobson, Maria R. Servedio.

**Visualization:** Elizabeth A. Hobson.

**Writing – original draft:** Emily H. DuVal.

**Writing – review & editing:** Emily H. DuVal, Courtney L. Fitzpatrick, Elizabeth A. Hobson, Maria R. Servedio.

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
