## [Editor Report · Decision Letter 0]

24 Apr 2023

Dear Dr DuVal, 

Thank you for submitting your manuscript entitled "Inferred Attractiveness and the evolutionary dynamics of mate choice in a social context" for consideration as a Research Article by PLOS Biology.

Your manuscript has now been evaluated by the PLOS Biology editorial staff, as well as by an academic editor with relevant expertise, and I'm writing to let you know that we would like to send your submission out for external peer review.

Once your full submission is complete, your paper will undergo a series of checks in preparation for peer review. After your manuscript has passed the checks it will be sent out for review. To provide the metadata for your submission, please Login to Editorial Manager (https://www.editorialmanager.com/pbiology) within two working days, i.e. by Apr 26 2023 11:59PM.

Kind regards,

Roli Roberts

Roland Roberts, PhD

Senior Editor

PLOS Biology

rroberts@plos.org

---

## [Decision Letter · Decision Letter 1]

16 Jun 2023

Dear Dr DuVal,

Thank you for your patience while your manuscript "Inferred Attractiveness and the evolutionary dynamics of mate choice in a social context" was peer-reviewed at PLOS Biology. It has now been evaluated by the PLOS Biology editors, an Academic Editor with relevant expertise, and by three independent reviewers. 

Based on the reviews, which are overwhelmingly positive, we are likely to accept this manuscript for publication, provided you satisfactorily address the remaining points raised by the reviewers and the following data and other policy-related requests.

IMPORTANT:

Please attend to the following:

a) Please could you change the Title to something more accessible and appealing to a broader readership? We suggest something like "Inferred attractiveness: females may acquire mating preferences by observing others’ choices in a social context"

b) Please attend to concerns raised by the three reviewers.

c) Please address my Data Policy requests below; specifically, we need you to supply the numerical values underlying Figs 1BCD, 2ABCD, 3AB, 4ABC, S1ABC, S2ABCD, S3i-xi, either as a supplementary data file or as a permanent DOI’d deposition. I note that you intend to deposit the Mathematica code in Dryad. Please could you clarify whether this will be sufficient to reproduce your Figures? If not, please supply additional data required. Please ensure that we have access to the Dryad deposition so that we can check it before accepting the manuscript for publication.

d) Please cite the location of the data clearly in all relevant main and supplementary Figure legends, e.g. “The data underlying this Figure can be found in S1 Data” or “The data underlying this Figure can be found in https://doi.org/10.5061/dryad.XXXXX”

We expect to receive your revised manuscript within two weeks. 

*Published Peer Review History*

*Press*

Sincerely,

Roli Roberts

Roland Roberts, PhD

Senior Editor,

rroberts@plos.org,

PLOS Biology

DATA POLICY:

Regardless of the method selected, please ensure that you provide the individual numerical values that underlie the summary data displayed in the following figure panels as they are essential for readers to assess your analysis and to reproduce it: Figs 1BCD, 2ABCD, 3AB, 4ABC, S1ABC, S2ABCD, S3i-xi. NOTE: the numerical data provided should include all replicates AND the way in which the plotted mean and errors were derived (it should not present only the mean/average values).

DATA NOT SHOWN?

REVIEWERS' COMMENTS:

Reviewer #1:

I reviewed manuscript PBIOLOGY-D-23-01008R1, "Inferred Attractiveness and the evolutionary dynamics of mate choice in a social context." The authors argue that, when individuals learn mate preferences by observing the mate choices of others, they may learn preferences for the most distinguishing traits of the chosen mates rather than for the traits that the original choosers used in their mate assessment. Using a mathematical model, the authors show that this can set up patterns of fluctuating and mostly negative frequency dependent mate preference at the population level, and that this can maintain multiple mate phenotypes in the population, even when those phenotypes are disfavored by viability selection. Most previous models predict that sexual selection due to learned or genetic mate preferences will squeeze variability out of mating traits, and it has been a long-standing paradox that variability in mating traits is common in nature. This paper offers a solution to that paradox. The authors' model is simple and compelling. The work falls into the class of results that we probably should have known but apparently didn't, which is the sweet spot for new theory. Moreover, the manuscript itself is clear and well-written. I believe this paper will change the way we think about sexual selection and set off a flurry of new theoretical and empirical work in this and related areas.

I have made a few minor suggestions below, but I want to be clear that these are just suggestions. I hope the authors can find something useful in them, but they are (as always!) free to take or leave them. 

Many of my suggestions stem from the observation that the methods section of this paper presents a lot of detail on how the model is coded. This may be useful for people who think in code, but may make the paper less accessible to theory savvy biologists who are not proficient programmers. For example, the authors tell us in where various pieces of information are stored in various vectors and matrices, but do we really need to know that? Would this not be exactly the same model if the authors had computed the frequencies of mating pairs using nested "for" loops rather than matrix operations? If so, then I think the authors might be better to present just the information that readers would need to understand the mathematical model in the methods, and to save the details of how they implemented that model for the annotated Mathematica notebook. I recognize that this is a matter of personal taste, and the authors should not have to do it or even respond to it. But, I encourage them to spend a few minutes thinking about what the paper might look like if the methods explained just the model, with less detail on how the model was implemented in their code.

Minor comments

Lines 143-144: The authors tell us that variability in the male traits in their model can be "lost" due to viability selection. This made me wonder about what causes fixation of genetic loci in this model. Does fixation occur stochastically despite negative frequency-dependent selection (eg, in generations when no females learn to prefer the rare male phenotype)? Or, is there some basin of attraction around a set of polymorphic states, and if the system wanders out of that basin does it proceed (in the absence of further stochasticity) deterministically to fixation of the male traits? I suspect one could answer this by studying the rates of change at the male trait loci when two of the male phenoptypes are vanishingly rare. I don't think the authors should spend days and pages figuring this out - if it is that much work it is probably a new project - but if they can answer it quickly and easily then it might help readers to understand the behavior of the system.

Figure 2: Does this figure illustrate the exact same simulations as Figure 1b? If so, that could be stated clearly.

Figure 3: In 3ai, do the bottom two panels show the same conditions (but different replicates) as the bottom two panels in 1b? And, in 3bi, do the left two panels show the same conditions (but different replicates) as the left two panels in 1b? If so, I would recommend either stating that clearly or removing the panels (since the goal of the figure is to show what happens when selection on the traits differs, and in these panels I think it does not differ). Also, should the headings of the right columns in 3ai and 3bi be "alpha = 7?" I guess there were 20 replicates for each combination of parameter values, but it would help readers if this were stated in the caption or legend.

Figure 4: When one trait is environmentally determined and viability selection on the genetic trait is not strong enough to overcome sexual selection, it makes intuitive sense to me that the frequency of the genetic trait gets "pinned" to the frequency of the environmentally determined trait. But, is there anything special about which allele gets pinned at low density? In the top right of 4a (ie, alpha=7, s=0.4), the favored allele is more common, which makes sense because viability selection is reasonably strong. In the bottom panels, where viability selection is weak, the disfavored allele is more common. But, does it have to be that way, or does it vary among replicates? These details might help readers more fully understand how the system behaves.

Line 352: Referring to the behavioral phenotypes as "loci" is confusing for me. I know the phenotypes have been coded like loci, but biologically they are not loci. I think this may be especially confusing for readers who are biologists and perhaps do not do a lot of coding - which is likely to be most of them, since I think this paper will have a broad readership. If I understand the model correctly, all of the important behavioral information about females can be captured by specifying the male phenotype they prefer. So, if it were my paper, I would describe females by just their genotype, their mate preference (as a single phenotype rather than three), and their age (even if, for computational convenience, I had tracked these things in a more complicated way).

Line 433: If I understand correctly, wbar here is not mean fitness, it is just mean survivorship between age classes. Fitness would include reproductive success. If that is true, then I recommend calling this mean survivorship and replacing w with some other variable, because many readers will expect w to be fitness. I also suspect that the model could be presented more simply if the authors did not try to index as in their code, but instead used, for example, m_ij(t) to indicate males born in generation t with alleles i and j at the TA and TB loci. Then I think they could define z_ij as the survivorship between age classes for males with genotype ij, and the proportion of males with genotype ij in the mating pool in generation t would just be m_ij(t-1) z_ij + m_ij(t-2) z_ij^2 divided by sum(m_xy(t-1) z_xy + m_xy(t-2) z_xy^2) for x and y in (1,2). Then, I do not think the authors would need equation 1 at all. I have not tried to rewrite the whole methods section using this notation, so maybe starting out this way would lead to trouble later, but if so I don't see it.

Line 574: "…while preserving any linkage disequilibrium that has built up between TA and TB." But, here, TB is an environmentally determined trait. So, what does a linkage disequilibrium represent in this case? Is there some assumption being made about offspring sharing the parental environment, and therefore the parental environmental trait? If not, I would think that no linkage disequilibrium could arise. In any case, it would help me as a reader to have the biological motivation for maintaining this linkage disequilibrium more clearly explained.

Reviewer #2:

[identifies himself as Gil G. Rosenthal]

This is an inspired and exciting model that breaks fundamental new ground in the way theoreticians have approached sexual selection. The model explicitly considers the complex interaction between public and private information, and shows that "inferred attractiveness" can explain much of the dynamism and variability observed in mate preferences in nature. This is a refreshing break from tweaks on the "good genes vs runaway" wrestling match that has dominated sexual selection theory. A pioneering model like this can only do so much in terms of exploring assumptions and parameter space. While the comparison to reference models is thorough and important, I think a little more discussion about the specificity and generality of the key assumptions would be helpful. To be clear, I think this is a rich model that goes beyond the unaided intuition to generate testable predictions, and I am not advocating more modeling....just perhaps a more nuanced approach to discussing the way the model does (and doesn't) map on to biology. I go through each of these in detail in the minor comments below.

1. It is refreshing that the preferences here do not have a simple genetic basis like in so many models. But to say that "genetically inherited preferences...lack widespread support" is misleading, particularly beyond vertebrates. A broad reading of the literature suggests that G X E effects are ubiquitous in shaping mating preferences much as they do other complex traits. Further, in many systems preferences are learned through early experience with relatives, so that there is a strong indirect genetic effect of parental phenotype on mating preferences; Servedio's work with Verzijden and others shows that this kind of vertically-transmitted preference is a more powerful driver of trait-preference coevolution than genetically specified traits and preferences. I understand that preferences aren't evolving in this model for the sake of simplicity, but part of "casting the female brain as the central selective agent" should be an acknowledgment that brains are evolving phenotypes like everything else.

2. Line 64. The paradox of the lek is not so much that there isn't any trait variation, but that the consequences of mating with one courter are no different from mating with another courter, and therefore there's no incentive to invest in mate choice. 

3. Line 77. The "rare male" effect - mate choice favoring rare phenotypes - has a long history in the literature (see e.g. O'Donald 1977 Nature, and Potter et al Science just this year). A little more thorough discussion of this phenomenon would be helpful, including an acknowledgement that choosers often prefer familiar phenotypes. See for example Alonzo and Sinervo, BES 2001 for an example where the fitness consequences of choosing rare males are variable for females. It's also worth mentioning that preferences for novel "bling" are generally anchored by stabilizing preferences for traits that are species-typical or that are predictive of sexual maturity or physical health (all of which can be explained by direct selection on female fitness rather than "good genes"). And if choosers prefer novel traits within this envelope, then one would expect rare mutations to be heavily favored. In the model, of course, there are only four possible trait combinations. How would a steady influx of novel trait variants change the model?

4. Figure 1b please use a third color rather than a different shade of blue for the second male trait.

5. line 328 choosers in nature exhibit variation not only in the strength but also the direction in which social experience affects preferences. Consider satin bowerbirds, where a high energy courtship display is attractive to older females and aversive to younger females, or the sister highland and sheepshead swordtails, which reject and favor familiar phenotypes, respectively. 

6. line 365 do we really need the term "phenogenotype"? There is a well established framework, employed by some of the authors of this paper, for talking about and modeling phenotypes that are shaped by genotype and environment. We can talk about different possible social effects on preference without calling them "alleles", which is unnecessarily confusing since preferences are evolving.

7. line 339. "choosers" and "courters" versus "females" and "males". It might facilitate comparison across the literature if terminology never changed, but on the other hand science and society have been pushed forward by changes in unnecessarily gendered language. For this particular model, the classic paradigm of coy females and promiscuous males is certainly the most tractable and interesting way to approach it, but it's hard to see how this model doesn't apply to male-limited systems where males are the choosers.

8. line 426 and 497. Age affects a ton of things in real populations. Old females are copied more than younger females, and old females rely less on public information (at least in guppies). Age-related changes and even reversals in preference (e.g. bowerbirds above) are not unusual. Female fecundity scales positively with age in many systems, as does male ornamentation and reproductive success. Given the importance of age structure to the dynamics of this model, this is worth mentioning. 

9. line 519 I am a bit befuddled as to how the renormalization of trait distributions was actually accomplished…maybe a supplementary figure would be helpful for this if I'm not the only one. I appreciate the importance of removing stochastic changes in trait frequencies to isolate the effects of preference evolution, but these stochastic changes…which by themselves change rare-male and copying dynamics…might change the game somehow, perhaps leading to fixation much more often than in the current model. 

Gil Rosenthal

Reviewer #3:

This paper presents an interesting and novel hypothesis of mate choice, developed as both a verbal and simulation model. "Inferred attractiveness" is a hypothesis based on mate choice copying and category learning that can explain fluctuations in preferences and sexual traits over time and contexts. I am excited to see this idea become further developed and tested. Suggestions for improvement of the manuscript follow, including major points first and then minor points line by line:

1. This hypothesis appears to be based on research in the cognitive sciences distinguishing classification and inference learning. These concepts should be explained in some detail—fleshing out the evidence and conceptual framework in references 36-38 so readers understand the empirical basis for the hypothesis; as written it does not seem as strongly justified as it could. I would recommend this come directly after the paragraph in which these references are introduced.

2. Along the same lines, it is not clear why the observing chooser should always choose the most rare phenotype that distinguishes the category "chosen mate" - further empirical or logical bases for this prediction would help convince the reader. 

3. Authors should speculate on the phylogenetic breadth of this mechanism. How many different species have been tested for this kind of category learning? Do the authors suppose this mechanism is as relevant for spiders and fish as it is for birds and people? 

4. L75-78—this sentence, and throughout, reads as if there is no innate or otherwise learned preference that "battles" with the inference. Please speculate as to how they might interact - or is it supposed that the "chooser" (Fig. 1a) is simply basing their preferences on what they learned, and so on, with no original preferences, only copies and inference?

5. L93—I don't think the authors mean "brighter" here—the chosen male looks less bright (more saturated red) than the unchosen.

6. L128-130—This sentence was hard to parse. "the trait of the chosen male that has the rarest" - is it the most rare allele of the trait, or the male with the rarest trait?

7. L156—Lines have no color in this figure.

8. L191-193—This sentence was hard to parse. What is consistent with what? I was similarly confused by 198-199. I'm not sure what represents a consistent pairing of facts.

9. L199-201—Again confused. I think this might help: "Weak sexual selection on one of two traits WHEN VIABILITY SELECTION ON THAT TRAIT IS STRONG results in loss of trait variation AT THAT LOCUS, regardless of the strength of sexual (or viability?) selection on the other trait."

10. Figure 3 - the levels of selection in the gray boxes are the same for a and b (ie constant sexual selection).

11. L241-242—Phrasing is awkward—suggest "In the case of preference for novelty, preference extremes are observed only in large groups."

L264—specify that "the 'selectively' favored genetic trait variant" is favored by viability or natural selection.

L324-327—this sentence did not make sense to me, not sure how to suggest a fix.

L415—I could have missed something, but what is "j mod"?

---

## [Editor Report · Decision Letter 2]

22 Jul 2023

Dear Dr DuVal,

Thank you for the submission of your revised Research Article "Inferred Attractiveness: a generalized mechanism for sexual selection that can maintain variation in traits and preferences over time" for publication in PLOS Biology. On behalf of my colleagues and the Academic Editor, Gail Patricelli, I'm pleased to say that we can in principle accept your manuscript for publication, provided you address any remaining formatting and reporting issues. These will be detailed in an email you should receive within 2-3 business days from our colleagues in the journal operations team; no action is required from you until then. Please note that we will not be able to formally accept your manuscript and schedule it for publication until you have completed any requested changes.

Sincerely, 

Roli Roberts

Senior Editor

PLOS Biology

rroberts@plos.org